# The Potential Role of Wild Suids in African Swine Fever Spread in Asia and the Pacific Region

**DOI:** 10.3390/v15010061

**Published:** 2022-12-24

**Authors:** Madalene Oberin, Alison Hillman, Michael P. Ward, Caitlin Holley, Simon Firestone, Brendan Cowled

**Affiliations:** 1Faculty of Veterinary and Agricultural Sciences, The University of Melbourne, Parkville, VIC 3010, Australia; 2Ausvet, Canberra, ACT 2617, Australia; 3School of Veterinary and Life Sciences, Murdoch University, Murdoch, WA 6150, Australia; 4Sydney School of Veterinary Science, The University of Sydney, Camden, NSW 2570, Australia; 5The World Organisation for Animal Health, Tokyo 113-8657, Japan

**Keywords:** African swine fever, Asia and the Pacific, wild suids, endemic pigs, feral pigs, population control

## Abstract

African swine fever (ASF) in Asia and the Pacific is currently dominated by ASF virus transmission within and between domestic pig populations. The contribution made by wild suids is currently not well understood; their distribution, density and susceptibility to the virus has raised concerns that their role in the epidemiology of ASF in the region might be underestimated. Whilst in the Republic of Korea wild suids are considered important in the spread and maintenance of ASF virus, there is an apparent underreporting to official sources of the disease in wild suids from other countires and regions. A review of the current literature, an analysis of the official reporting resources and a survey of the World Organisation of Animal Health Member delegates in Asia and the Pacific were used to assess the potential role of wild suids in ASF outbreaks, and also to gain insight into what ASF management or control strategies are currently implemented for wild suids. Applying appropriate population control and management strategies can be increased in some areas, especially to assist in the conservation of endangered endemic wild suids in this region.

## 1. Introduction

There is an ongoing global panzootic causing high mortality in certain suids as a result of African Swine Fever virus (ASFV) infection. ASFV is a large, double-stranded DNA virus from the *Asfarviridae* family, and is enveloped with genomic DNA of approximately 170–193 kb [1]. The current spread involves predominantly the transmission of genotype II strains of the ‘Georgia 2007’ type virus [2]. Some attenuation has been observed in Estonia, but most strains observed during the current ASF panzootic are highly virulent [3,4]. There is currently no vaccination available for use in controlling the disease, although there has been an increasing research focus on this area [5]. ASFV has many transmission routes; these include direct contact with an infected pig or pig carcass, scavenging of infected carcasses or pork products from infected animals, contact with fomites contaminated with blood, faeces, urine or saliva from infected pigs (including bedding, feed, equipment, clothes and footwear, and vehicles), and spread by *Ornithodoros* spp. ticks (particularly *O. moubata*) [6]. ASFV is robust to degradation in pork products and in the environment [7]. There are several forms of ASF disease: acute, which leads to death of up to 100% of infected pigs, typically after 6–13 days; subacute, where mortality rates are lower (30–70%) and clinical signs can be exhibited for long periods of time; and chronic, in which mortality is low and a small number of affected individuals may become virus carriers for life with periodic viraemia [8,9,10]. However, some authors contend that there is insufficient evidence of a subclinical carrier state [11,12]. Eurasian wild boar and feral pigs (both *Sus scrofa*) are highly susceptible to disease, with clinical signs and mortality rates equivalent to those seen in domestic pigs (also *S. scrofa*) [13]. The impacts of ASF outbreaks have had a cascading effect, threatening domestic populations, the livelihood of farmers (particularly smallholders), the food supply chain and ultimately food security [14]. The panzootic might also put further pressure on the population viability of already threatened endemic wild suid species [15,16].

In terms of the broad geographical progression of the current panzootic, suid populations in the Republic of Georgia and the Russian Federation were first exposed to ASFV in 2007. This probably occurred through improper disposal of contaminated pork meat from a ship at Poti docks [17], which led to widespread infections in European domestic pigs and wild boar from 2014 onwards [18,19]. During August 2018, African swine fever was first reported in Asia, China, then spread rapidly across the North-East in domestic pigs; it then spread to the South-East, likely due to local suid movements and ineffective biosecurity [20]. It has since spread to other areas throughout Asia, and into Papua New Guinea, which has been the only region in the Pacific to have experienced an outbreak. Geographical, ecological and epidemiological evidence indicates that ASFV transmission is multifaceted and varies across different regions of the world, with differences in the roles that wild and domestic pigs play in virus maintenance and spread [21]. In many European outbreaks, particularly in western Europe, domestic pig infections have been primarily due to wild boar sources [8,22,23,24]. However, in other areas there has been evidence of both wild and domestic pig populations contributing to ASFV spread. For example, in Romania, outbreaks in domestic pigs were associated with proximity to outbreaks in wild boar and wild boar abundance [24]. Similarly, in some Mediterranean countries, it was observed that where the disease is actively circulating in domestic pig populations, there were also high densities of wild boar and contact between wild boar and free-ranging pigs, suggesting that wild boar have a role in the spread of the virus [23]. In contrast, in other regions in Eurasia, outbreaks have been initiated within the domestic population cycle, wild boar being of secondary importance to transmission (although regular spillover events and sometimes virus transmission were inferred). For example, in the Russian Federation and Caucasus, ASFV transmission is associated with the movement of live domestic pigs and pork products, or with poor biosecurity in smallholder pig production [25]. It may be inferred that where wild boar outbreaks have been reported in regions of Asia, the direction of transmission is more commonly domestic pigs to wild boar than vice versa [25,26].

More is known about the epidemiology of ASF in wild suids in Europe than in Asia and the Pacific region, as more surveillance, research and epidemiological analyses have been conducted [8,24,25,27]. This knowledge can assist in understanding the epidemiology of ASF and suitable management interventions within Asia and the Pacific contexts, where evidence is still limited. For example, in western Europe, carcasses appear critical to the overwintering of ASFV in a region, whereas this might be less of a concern in particular regions of Asia and the Pacific that have warm tropical climates [28,29]. The large geographical region of Asia Pacific is in a unique position, containing twelve species of locally endemic wild suids which are unevenly distributed across the Asian part of this region. These species are generally in population decline, and most have a sub-optimal conservation status such as threatened or endangered [30]. The exception is the common and widespread *S. scrofa*, which occurs as domestic pigs, Eurasian wild boar or feral pigs. The locally endemic pig species contribute substantially to the diversity of the Suidae family globally, and are an important conservation resource. In contrast, in the Pacific and Australasian regions wild suids are introduced *Sus scrofa* (feral pigs), which although an invasive pest species [31,32] can be highly valued as a food and cultural resource. As all Suidae species are believed to be susceptible to ASFV infection [16], this creates difficulty in balancing ASF management and control scenarios, whilst respecting the diversities of cultures, food security and conservational importance of wild suids [23,33].

The objectives of this paper are to review the role of wild suids in ASF spread across the Asia and Pacific region, including the ecology and distribution of wild *Sus scrofa* and endemic wild suids, ASF reports, transmission pathways, and control measures in place for wild suids. We address these objectives by a review of the literature, an assessment of national surveillance data analysis and an expert opinion survey.

## 2. Materials and Methods

This paper includes a scoping review of published literature and reports from official databases; and a follow-up survey of World Organisation for Animal Health (WOAH) delegates and representatives in Asia and the Pacific.

### 2.1. World Organisation for Animal Health Wild Suid in the Asia Pacific Region Project

This review was developed following a broader report developed for the WOAH Asia Pacific regional office on the role of wild suids in transmission of the ASF panzootic in the Asia and the Pacific region. The full project report was published by the WOAH [34], and pertinent parts were refocused to produce this paper. The project team worked with a WOAH expert advisory group from the region.

### 2.2. Study Area and Definitions

The study area is defined as the 32 WOAH Member States within the Asia and Pacific region (Table 1).

In this paper, ‘wild suids’ are considered unmanaged suid populations, including wild or feral populations of *Sus scrofa*, other wild suidae and hybrids, whilst domestic pigs refer to managed populations (further defined in Appendix A).

### 2.3. Literature Review and Official Reports

For this review, a hybrid qualitative−quantitative approach was used to gather recent and available information on the potential role wild suids may have on the ASF situation in Asia and the Pacific. This was achieved by searching a combination of online literature databases (Web of Science, Medline and PubMed Central), using key terms such as; “African swine fever” OR “African swine fever virus” OR “ASF” OR “ASFV” AND “wild pig” OR “feral pig” OR “wild boar” OR “endemic pig” OR “*Sus scrofa*” OR “*Sus*” OR “*Babyrousa*” OR “*Porcula*” AND “outbreak” OR “transmission” OR “spread” OR “susceptibility” AND “Asia” OR “Pacific” OR “Oceania”. In addition to searching grey literature and through formal meetings with the Food and Agriculture Organisation of the United Nations (FAO)/WOAH Global Framework for the progressive control of Transboundary Animal Diseases (GF-TADs) standing group of experts on ASF [35,36]. The World Animal Health Information System (WAHIS), FAO and IUCN databases were used to gather quantitative data on the spread of the disease, reported outbreaks and cases, and the wild suid species distribution and conservation status across the study extent [37,38,39]. Specifically, the ‘disease situation’, ‘qualitative data’, and ‘control measures’ dashboards were used from WAHIS and data were collected from the online system, with the last search for this paper being conducted on the 7 August 2022 [40,41,42].

### 2.4. Survey

This study was undertaken following protocols approved by the University of Melbourne Human Ethics Committee (application number 2021-23073-23872-3). The online survey tool Qualtrics [43] was used to design, create and distribute the questionnaire, which comprised mostly closed questions (Appendix A, Appendix A). The survey was distributed in October 2021 by email to WOAH Member delegates within Asia and the Pacific via the WOAH regional representative network. It remained open for approximately 6 weeks. In instances in which more than one delegate from a Member provided a response to the survey, where possible their results were amalgamated to represent a single response. If there were contradictory answers provided by multiple respondents from the same Member, the answer was amalgamated with preference given to ‘yes’ answers—for example, if there was an ‘unsure’ or ‘no’, and a ‘yes’ response for the same question, ‘yes’ was used as the final answer.

Descriptive statistical outputs from the Qualtrics survey tool were examined. The responding Members were categorised as; low, lower middle, upper middle or high income status, based on the latest World Bank data [44]. Throughout the analysis, *Sus scrofa* was categorized as ‘wild boar’ if it was endemic in the Member State, otherwise as ‘feral pigs’ if introduced and naturalized.

## 3. Results

### 3.1. Ecology and Distribution of Wild Sus scrofa in the Region

*Sus scrofa* is endemic in the Sino-Japanese and Oriental zoogeographic regions of Asia and the Pacific. In contrast, they are introduced in the Oceania and Australian zoogeographic regions, where they inhabit almost all islands, including Australia, New Zealand, and Papua New Guinea. In many of these locations, the populations are a hybrid mix of introduced *Sus scrofa* and domestic pigs [32].

Broadly, *Sus scrofa* is widely distributed and abundant across Asia and the Pacific [16,37,45]. However, specific information on the distribution, density and ecology of the unmanaged populations of *Sus scrofa* within the Asia and the Pacific region tends to be limited, with few exceptions (e.g., Australia [46,47]). Recent studies have innovatively used geo-mapping systems parameterised with the estimated climatic and topographic tolerance limits of *Sus scrofa* to broadly assess the distribution of pigs based on available habitats within Eurasia [48,49]. These studies concluded that ecological patterns have a major role on wild suid distribution and density in the Asia and the Pacific region.

Wild *Sus scrofa* are very adaptable and so found in various subalpine, temperate, subtropical and tropical habitats in the region, including riparian areas, semi-desert areas, rainforests, woodlands, grasslands and reed jungles [32,37,50,51]. They are typically found near thick vegetation, and if the area is warm or dry, then close to a water source [50]. This can make wild *Sus scrofa* difficult to observe and survey, and thus also make management and surveillance challenging.

Wild *Sus scrofa* are opportunistic omnivores: their diet varies depending on location and food availability. It is usually predominantly plant material, with some animal components such as carrion (including pig carcasses), small mammals and livestock (e.g., sheep) [37,52,53,54]. As well as cannibalistic scavenging, *Sus scrofa* may show behavioural interest in conspecific carcasses, including investigating the soil next to and under them for invertebrates and thereby making direct contact with the carcass [55]. Both of these factors represent a considerable risk in the transmission of ASFV to *Sus scrofa* in some environments, as carcasses can remain infectious for substantial periods of time under favourable (cold) conditions, with experimental conditions confirming ASFV detection ranging from 3 months to over 2 years [56,57,58].

Social behaviour and group size of ‘sounders’—the matrilineal (female) groups in which wild suids live—can also considerably affect disease transmission. *Sus scrofa is* non-territorial and social, with overlapping home ranges and interactions between separate sounders or larger herds of wild suids. Sounders generally comprise up to 50 individuals; however, when water is scarce and sounders aggregate at available water sources larger groups of up to a hundred pigs have been observed [32,59]. Population densities of sounders can be greater than 20 pigs/km^2^, with large home ranges of up to 30 km^2^, depending on available food resources (where food resources are scarce—for example, during colder or drier seasons—home ranges may be relatively increased) [60,61,62,63]. Proximity to water sources is important in hot environments for thermoregulation [50], and wild suid sociability leads to close contact at such aggregation points. Given that *Sus scrofa* infected with ASFV are thought to specifically seek cool, moist and sheltered environments (including water-related areas) to ease clinical signs [64], there is a high potential for the spread of infections within and between separate groups of pigs at such locations.

In Asia and the Pacific region, domestic pigs may be free-ranging, semi-free ranging or housed, and may be located near wild suid populations. Interaction and contact have been observed globally between free-ranging or housed domestic pigs and wild suids in Europe [23,65,66] and the Asia and the Pacific region [67,68]. These interactions may be an important factor in the spread of ASFV to wild suid populations.

There were 35 responses to the survey, representing 27 different WOAH Members within Asia and the Pacific; not all Members responded to every question. The most common species present was *Sus scrofa* (wild boar or feral pig) (96%, *n* = 26/27 Members). Whilst many Members knew they had wild suids present, many (67%, *n* = 18/27) noted there was no information on the distribution or density of wild suids, or left these sections blank on the survey, suggesting a lack of knowledge.

### 3.2. Ecology of Endemic Wild Suids and Their Potential Role in African Swine Fever Transmission

There are 12 species of endemic wild suid in Asia. These include the abundant *Sus Scrofa*, 10 species endemic to South-east Asia (including seven *Sus* spp. and three *Babyrousa* spp.), and *Porcula salvinia* (endemic to North-East India). These species contribute substantially to the diversity of Suidae species globally and are an important conservation resource. All these endemic species, except for *Sus scrofa*, are declining in population distribution and abundance (acknowledging a lack of recent data available for many species; Table 2). Their conservation status varies from near threatened to critically endangered due to various processes (including habitat loss). Within these wild suid species, some features may contribute to the epidemiology of ASF. For example, bearded pigs (*S. barbatus*) can migrate vast distances to forage for fruits, which could facilitate the spread of ASFV if pigs are incubating the virus. However, the incubation period and whether there are carrier states for ASFV in bearded pigs is unknown.

### 3.3. African Swine Fever in Wild Suids

African swine fever in wild suids has been primarily documented in the Asian countries of the region (Figure 1). Anecdotally, ASFV detections in wild suids have regularly been judged to be secondary to domestic pig outbreaks, having been confirmed several months after domestic pig outbreaks in the same area [39,80]. For example, in Vietnam ASFV was confirmed in domestic pigs in February 2019, and then in wild boar in December 2019 [45]. Meanwhile, Cambodia has yet to report ASFV in wild suids despite confirmation in domestic pigs in March 2019 [45]. It has been suggested that the spread of ASFV by wild suids is underestimated in Asia [22,81].

Cases of ASF have been documented in bearded pigs in Borneo [71,72,73] and Philippine warty pigs [75]. In the case of Philippine warty pigs, it was specifically noted that the disease appeared similar to that in domestic pigs [75]. ASFV was not reported in the other endemic species, which may be attributed to limited surveillance activities within the regions or a lack of transmission to these species (Table 2).

The susceptibility of the other wild suid species in which ASF has not yet been reported is poorly understood, though plausible [8,16,23,33]. Pigs of the genus *Sus* (bearded pigs and warty pigs) are predicted to be similarly susceptible to infection and disease as other *Sus* species and subspecies; it is not known if susceptibility or ASFV virulence differs in wild suids of other genera (*Babyrousa* spp. and *Porcula salvania*) [82].

The reporting to WAHIS (as of 7 August 2022) of ASF in wild suids is limited [38]. Reports have been submitted by four WOAH Members in the region—China, the Republic of Korea, Laos, and Malaysia (Table 3, Appendix A). However, it was found in the literature that 9 Members have had ASF outbreaks in wild suids (Table 4; [39]), an increase compared to the WAHIS database.

#### African Swine Fever Transmission Pathways Reported in the Survey

The most frequent reported routes of ASFV transmission between domestic and wild suid populations are direct contact or scavenging of infected carcasses; transmission may also occur through wild suids having access to swill and contact with fomites and effluent from pig production (Table 5).

Of the participating WOAH Members, ASF in wild suids has been detected predominantly in wild boar/feral pigs (*Sus scrofa*) (39%, *n* = 9/23). It has also been detected in bearded pigs (*Sus barbatus*) and Philippine warty pigs (*Sus philippensis*) by two separate members. The question regarding the transmission of ASFV was left unanswered by many respondents possibly due to limited surveillance in wild suids in the area, however the small quantity of answers reported transmission occurring in both directions between wild and domestic pigs (Table 5).

### 3.4. Control Measures in Place for African Swine Fever

There is currently no viable vaccine or treatment available for ASF, placing reliance on prevention and traditional control strategies to protect pig populations [101]. The application of appropriate control measures depends on numerous factors, such as the location, type of pig production, movements of animals, and biosecurity strategies in place [102]. Almost all WOAH Members (30/32) in Asia and the Pacific have reported to WAHIS at least one control measure in place for ASF, with general surveillance and disease notifications being the most common (Table 6). However, in the survey 17/23 (73.9%) Members reported the use of specific control or prevention strategies for ASF in wild suids, which were not represented in the WAHIS data. Overall, a variety of control methods were used, more frequently in high income Members than low-to-middle income Members (Table 7). There were fewer responses within the survey (*n* = 17) compared to WAHIS *(n* = 30), with surveillance being one of the most used control measures. Population control/culling tools were reported by 13% (*n* = 3/23) of responding Members. All three used hunting (i.e., shooting on the ground and trapping) as a part of this strategy; one Member also used poison baiting and aerial shooting.

### 3.5. Correlation between Survey-Reported Control Measures and the Implementation of Other Prevention Strategies

#### 3.5.1. Biosecurity, Pig Production Type and Wild Species Present

Domestic pig farming of *Sus scrofa* with poor biosecurity represents a considerable risk for ASFV transmission, especially where wild suids are in close proximity to domestic pigs. The majority of Asia and Pacific Members conduct domestic pig farming (72%, *n* = 18/25), of which small-scale (*n* = 15) and medium-scale (*n* = 14) production were reported in the survey to be the two most common systems (Appendix A). Large-scale production was also reported (*n* = 11). The use of small-scale production systems occurred mostly in developing Members (*n* = 11), and 54.5% (*n* = 6) use free ranging/scavenging systems, of which half also had wild suids present. The control measures implemented by these developing Members were minimal (Table 7). For example, the survey found biosecurity was only used in 18% (*n* = 2/11) and fencing was not used by any of the developing Members with free-ranging systems that were reported. Overall, biosecurity practices were implemented in mostly developed Members (78%), and not utilized in any Members where endemic wild suid populations were present.

#### 3.5.2. Wild Suid Carcass Disposal and Climatic Conditions

The disposal of dead wild suid carcasses was reported to be used in six different Members, of which four are considered to commonly experience winters below −5 °C (and during summer, these four Members rarely experience temperatures above 25 °C). The very low temperatures are ideal for carcass preservation in the environment, thus removing dead wild suid carcasses may substantially limit opportunities for transmission of ASFV in these countries. In comparison, the two other Members that reported removing dead pig carcasses rarely experience such extreme cold weather, with temperatures above 30 °C common during summer. It was assumed that reporting the carcass disposals of endemic wild suid populations would be too infrequent to include in the survey due to their low population densities.

#### 3.5.3. Border Quarantine and Land Type

Border quarantine strategies were reported to be in place in six different Members, of which five represent land that is classified as islands. Border quarantine on islands is generally easier to implement, and incursions and imports of wild suids or infected products are easier to manage, than for Members separated by land borders.

#### 3.5.4. Wild Suid Management Strategies and *Sus scrofa* Status

The majority of responding Members reported managed hunting strategies for wild suids (52%, *n* = 12/23). Reasons for hunting were primarily for food (78%, *n* = 7/9), and game (recreational/sporting) (67%, *n* = 6/7). In eleven of these responding Members, *Sus scrofa* is considered a native species (wild boar); whereas where *Sus scrofa* are an introduced species (feral pigs) managed hunting was reported by 8% (*n* = 1/12) of Members. Thus, native *Sus scrofa* regions are more likely to have a managed hunting system, compared to where *Sus scrofa* is introduced.

#### 3.5.5. Legislation and Regulations

The regulations or legislation regarding wild suids varied across the different categories (conservation, control of ASF and hunting; Table 8). Within these categories, laws about the hunting of wild suids are of the highest concern, with ASF control laws being second.

## 4. Discussion

### 4.1. Occurrence of African Swine Fever in Wild Suids

African swine fever outbreaks have primarily been reported in domestic pig populations in Asia and the Pacific, and the spread of the disease is predicted to continue [39,103]. Although generally presumed to be less of a concern compared to other regions of the world, the occurrence and distribution of ASF in wild suids in Asia and the Pacific is uncertain, given the scarcity of current evidence. The absence of, or limited official reporting of ASF cases in wild suids cannot be presumed to indicate the absence of infection: many Members have limited active surveillance activities in wild suid populations, or surveillance activities are restricted to general surveillance, which might be insensitive in detection and confirmation of the infection in wild suids [22,104]. For example, there might be logistical reasons for an inability to detect ASF in certain ecosystems, or difficulties relating to local authorities and resources [22]. Given the frequent detections of ASFV in domestic pigs across the region, active surveillance activities targeting wild populations in at-risk regions would be beneficial.

### 4.2. Wild Suids Role in Disease Spread

Inferences as to the role of wild suids spreading ASF in Asia and the Pacific are currently inferred based on observations from other geographic regions of the world where enhanced surveillance, research and analyses have been undertaken [8,24,25,27]. This information is regularly used as a substitute for the lack of data within Asia and the Pacific, but such inferences must be made with caution. It is evident that *Sus scrofa* can be found in abundance throughout Asia and the Pacific, and thus have the potential to spread and maintain ASFV in the region. In comparison and irrespective of uncertainties in susceptibility to infection and disease transmission dynamics, the 11 non-*Sus scrofa* endemic pig species are unlikely to have an important role in ASF epidemiology in the region, as their populations are small and have limited distributions.

However, more broadly the distribution of wild suid species and their population densities across the region is uncertain, and this information should be investigated further to assess the role of wild suids in spreading the disease with greater confidence, given the potential implications for ASFV transmission dynamics. Further, ecological attributes may affect the maintenance of infection in wild suid populations, and thus influence the likelihood of transmission from these populations to domestic pigs. For example, geographic locations and times of year in which ambient temperatures are relatively high may influence the likelihood of virus transmission through the relatively rapid decomposition of wild suid carcasses and negative impacts on the virus’ viability in the environment.

The risk of transmission of infection from wild suids to domestic pigs is also influenced by farming methods [105]. In the Asia and the Pacific region, many Members have a heavy reliance on small-scale production systems that typically involve free-ranging/roaming methods.

### 4.3. Potential Improvements on Control/Management Strategies

The impacts of ASF justify proactive strategies in the prevention and management of outbreaks. The suitability of different strategies might vary by epidemiological context across the Asia and the Pacific region. For example, in climates where wild suid carcasses could naturally decompose at a fast rate reducing the potential for ASFV transmission, resources should be targeted at other control measures that are likely to be more effective, such as border quarantine or on-farm biosecurity interventions where transmission risks are elevated. Further research is required to inform strategic approaches that best cater for Members’ own situations and specific requirements.

Areas of Asia and the Pacific region that are currently experiencing ASF outbreaks can undertake interventions in wild suid populations to reduce disease transmission (e.g., by decreasing wild *Sus scrofa* population density with control tools if appropriate such as trapping, aerial shooting, poison baiting and intensive hunting) and to prepare for a possible vaccine (through the development of bait delivery strategies, including relevant ecological research). On-farm biosecurity improvements, such as education about infectious diseases, and establishment of village-level biosecurity practices regarding isolation of moved pigs and confining/penning pigs, might also reduce the risk of transmission between domestic and wild suids.

For areas of the Asia and Pacific region that have yet to detect ASF, being proactive by implementing prevention, detection and response strategies can assist in remaining free or reducing the impact of an outbreak. For example, prevention via border quarantine (which is especially relevant to islands), reducing time to detection of incursions by improving general surveillance activities in wild and domestic pig populations, and implementing appropriate response frameworks to manage ASFV transmission to, from and within wild suid populations once it is detected.

Importantly, ASF may have catastrophic population-level impacts on the 11 non-*Sus scrofa* endemic wild suid species, threatening populations with local extirpation or extinction. Hence, increased protection through breeding and conservation measures should be considered where these species are endemic. For example, risk assessment with appropriate mitigation strategies to protect important populations of endemic wild suids or captive breeding programs.

## 5. Conclusions

The role of wild suids in the epidemiology of the disease in Asia and the Pacific is poorly understood, though wild *Sus scrofa* has potential to contribute to the spread of disease. Nevertheless, population interventions in wild suid populations may be necessary to control disease outbreaks, and actions to prevent infection in threatened species populations are important considerations. Whether ASFV can survive for long periods in wild suid carcasses in warmer regions of Asia and the Pacific is an information gap requiring research.

## Figures and Tables

**Figure 1 viruses-15-00061-f001:**
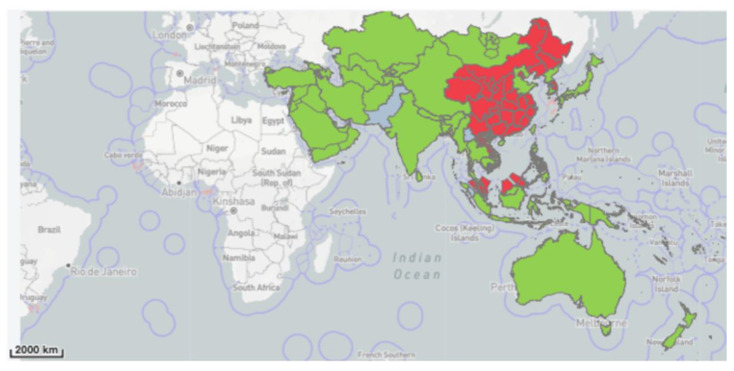
African Swine Fever cases in wild suids reported to the World Animal Health Information System, red shaded regions representing African swine fever is present and green representing absent [41].

**Table 1 viruses-15-00061-t001:** World Organisation for Animal Health (WOAH) Members within the Asia and the Pacific Regions.

WOAH Members for the Asia and the Pacific Region			
Australia	India	Maldives	Papua New Guinea
Bangladesh	Indonesia	Micronesia (Federated States of)	Philippines
Bhutan	Iran	Mongolia	Singapore
Brunei	Japan	Myanmar	Sri Lanka
Cambodia	Korea (Democratic People’s Republic of)	Nepal	Thailand
China (People’s Republic of)	Korea (Republic of)	New Caledonia	Timor-Leste
Chinese Taipei	Laos	New Zealand	Vanuatu
Fiji	Malaysia	Pakistan	Vietnam

**Table 2 viruses-15-00061-t002:** Summary of wild suid species, current African swine fever (ASF) status, distribution, and population size (data mostly collected from the International Union for Conservation of Nature [15]) in the Asia Pacific region.

Species	ASF Detected	Distribution	Population Size (Estimate)
Wild boar/Feral pig (*Sus Scrofa*)	Yes [13]	Widely distributed across Asian countries in the oriental and Sino-Japanese zoogeographic regions ^1^	Abundant throughout the region
Sulawesi babirusa (*Babyrousa celebensis*)	No	Indonesia	9999 [69]
Hairy babirusa (*B. babyrussa*)	No	Indonesia	No recent data available ^2^
Togian Islands babirusa (*B. togeanensis*)	No	Indonesia	1000 [70]
Bearded pig (*Sus barbatus*)	Yes [71,72,73]	Indonesia, Brunei, Malaysia ^3^	No recent data available
Javan warty/Bawean warty pig (*S. verrucosus*)	No	Indonesia	*S. v. blouchi*: 172–377 [74] ^4^
Sulawesi warty pig (*S. celebensis*)	No	Indonesia	No recent data available
Philippine warty pig (*S. philippensis*)	Yes [75]	Philippines	No recent data available
Mindoro (oliver’s) warty pig (*S. oliveri*)	No	Philippines	No recent data available
Palawan bearded pig (*S. ahoenobarbus*)	No	Philippines	No recent data available
Visayan warty pig (*S. cebifrons*)	No	Philippines	No recent data available
Pygmy hog (*Porcula salvania*)	No	India, Bhutan ^5^	100–250 [76]

^1^ As defined in Holt et al. (2013) [77]. ^2^ In 2000, an estimate was made of 4000 individuals [78]. ^3^ Extinct in Singapore, possibly extinct in the Philippines [79]. ^4^ No recent data are available for the species more broadly. ^5^ Presence in Bhutan is uncertain [76].

**Table 3 viruses-15-00061-t003:** Reports of African swine fever in World Animal Health Information System in wild suids in World Organisation for Animal Health Members of Asia and the Pacific.

Member	Year	Species	New Outbreaks	Cases	Killed and Disposed of	Deaths
China (People’s Republic of)	2018–2020	Wild boar	4	316	310	304
Laos	2019	Suidae (unidentified)	2	6	0	6
Malaysia	2021	Wild boar	50	115	0	115
		Bearded pigs	2	13	0	13
			2	5	-	5
Korea (the Republic of)	2019–2022	Wild boar	2047	2856	227	2350
	2021	Suidae (unidentified)	4	4	0	4
	2021	-	1	1	0	1

**Table 4 viruses-15-00061-t004:** Reported outbreaks of African swine fever by World Organisation for Animal Health members in Asia-Pacific from 2018 to present (7 August 2022).

Member	Outbreak Start Date, Status	Classification of Infected *Sus scrofa*	Details of Disease Spread
Bhutan [83,84]	6 May 2021 (ongoing)	Wild and domestic	Detected in free-roaming pigs, then to semi-commercial farm.
Cambodia [85]	March 2019 (resolved),	Domestic	Detected in backyard pigs, from unregulated importing of pork.
China (People’s Rep. of) [86,87,88]	1 August 2018 (ongoing)	Domestic and wild	Likely domestic contaminated wild, hypothesised spread due to tick-to-pig transmission.
India [85,89]	26 January 2020	Domestic and wild	Detected in domestic pigs then in dead wild boars. Predicted to be from wild boar-habitat cycle.
Indonesia [16,85,90,91]	17 December 2019	Domestic and wild	Source is unknown, spread by animal–human-vehicle-animal.
Korea (Dem. People’s Rep. of) [92]	23 May 2019	Domestic	Detected in Chagang-do (border with China)
Korea (Rep. of) [93]	17 September 2019	Domestic and wild	Predicted to spread from domestic to wild boar by Anthropogenic interactions.
Laos [45,85]	20 June 2020 (resolved)	Domestic and wild	Spread from domestic to wild boar due to free-ranging farming styles.
Malaysia [94]	8 February 2021 (ongoing)	Wild and domestic	First case in domestic pigs was triggered after wild boar case.
Mongolia [95]	10 January 2019 (resolved)	Domestic	Likely spread due to swill feeding.
Myanmar [96]	14 August 2019 (ongoing)	Domestic	Detected in a farm due to pigs dying.
Nepal [97]	16 May 2022 (ongoing)	Domestic	Likely spread due to swill feeding
Papua New Guinea [98]	5 March 2020 (ongoing)	Domestic	Unknown/ inconclusive of source or origin, spread by illegal imports of infected pork products and scavenging.
Philippines [85,99]	25 July 2019 (ongoing)	Domestic and wild	Suspected to have spread after a resident imported a wild boar.
Thailand [84]	January 2022 (ongoing)	Domestic	Detected in companion pigs and during slaughtering.
Timor-Leste [100]	9 September 2019 (ongoing)	Domestic	Unknown source, likely due to transporting infected pigs.
Vietnam [45,85]	1 February 2019 (ongoing)	Domestic and wild	Detected in domestic pigs then wild boar. Spread is likely due to farming method and spillover by domestic pigs

**Table 5 viruses-15-00061-t005:** African swine fever virus transmission routes in wild suids reported by 5 Members within regions of Asia and the Pacific.

**Direction of transmission nominated in the survey** **responses**	Domestic—Wild (60%, *n* = 3/5) Wild—Wild (60%, *n* = 3/5) Wild—Domestic (60%, *n* = 3/5) Unsure (40%, *n* = 2/5)
**Transmission** **mechanism nominated in the survey responses**	Direct contact (pig-to-pig) (*n* = 5) Direct contact with infected dead pig carcass (*n* = 3) Scavenging of food/waste from domestic pig farms (*n* = 3) Indirect contact (*n* = 3) Spread via pig effluent from domestic piggery (*n* = 1) Spread via pig products (i.e., pork) (*n =* 1)

**Table 6 viruses-15-00061-t006:** Control measures reported in place for ASF in wild suids, per WAHIS [40] reporting and FAO [39], in the Asia Pacific region.

Income Status *	Member	General Surveillance	Targeted Surveillance	Disease Notification	Monitoring	Zoning	Control of Wildlife Reservoirs	Control of Vectors
High income	Australia	Yes	Yes	Yes	Yes	-	-	-
	Brunei	Yes	-	Yes	-	-	-	-
	Chinese Taipei	-	-	Yes	-	-	-	-
	Japan	Yes	-	Yes	Yes	-	Yes	-
	Korea (Republic of)	Yes	-	-	-	Yes	Yes	-
	New Caledonia	-	-	Yes	-	-	-	-
	New Zealand	Yes	-	Yes	-	-	-	-
	Singapore	Yes	-	Yes	Yes	-	-	-
Upper middle	China (People’s Republic of)	Yes	Yes	Yes	Yes	Yes	Yes	Yes
	Fiji	Yes	-	Yes	-	-	-	-
	Malaysia	Yes	Yes	Yes	Yes	-	Yes	-
	Maldives	-	-	Yes	-	-	-	-
	Thailand	-	-	Yes	-	-	-	-
Lower middle	Bangladesh	Yes	-	-	-	-	-	-
	Bhutan	Yes	Yes	Yes				
	Cambodia	-	-	-	-	-	-	-
	India	-	-	Yes	-	-	-	-
	Indonesia	Yes	-	Yes	Yes	-	-	-
	Iran	-	-	-	-	-	-	-
	Lao	Yes	Yes	-	-	Yes	-	-
	Micronesia (Federated States of)	-	-	Yes	-	-	-	-
	Mongolia	Yes	-	-	-	-	-	-
	Myanmar	Yes						
	Nepal	Yes	-	-	-	-	-	-
	Pakistan	-	-	-	-	-	-	-
	Papua New Guinea	Yes	-	-	-	Yes	-	-
	Philippines	Yes	-	Yes	-	Yes	-	-
	Sri Lanka	-	-	-	-	-	-	-
	Timor-Leste	-	-	-	-	-	-	-
	Vanuatu	-	-	Yes	-	-	-	-
	Vietnam	Yes	-	-	-	-	-	-
Low	Korea (Democratic People’s Republic of)	-	-	Yes	-	-	-	-
	TOTAL	19	5	19	6	5	4	1

* Income status as per the World Data [44].

**Table 7 viruses-15-00061-t007:** Proportion of different control methods for *Sus scrofa* applied throughout Asia and the Pacific reported in the survey.

	Fencing	Zoning	Biosecurity	Surveillance	Carcass Disposal	Vector Control	Culling/Population Control	Border Quarantine
*Sus Scrofa*	24% (*n* = 4/17)	18% (*n* = 3/17)	53% (*n* = 9/17)	53% (*n* = 9/17)	29% (*n* = 5/17)	-	18% (*n* = 3/17)	35% (*n* = 6/17)
Proportion implemented in:								
High-income Members	75% (*n* = 3/4)	100% (*n* = 3/3)	67% (*n* = 6/9)	44% (*n* = 4/9)	80% (*n* = 4/5)	-	100% (*n* = 3/3)	83% (*n* = 5/6)
Upper Middle-income Members	25% (*n* = 1/4)	-	11% (*n* = 1/9)	-	-	-		17% (*n* = 1/6)
Lower middle-income Members	-	-	22% (*n* = 2/9)	56% (*n* = 5/9)	20% (*n* = 1/5)	-		
Low-income Members	-	-	-	-	-	-	-	

**Table 8 viruses-15-00061-t008:** The proportion of Members implementing regulations or legislation for conservation, African Swine Fever (ASF) control and hunting of wild suids.

	Conservation of Wild Suids	For ASF Control	For Hunting Wild Suids
Yes	30% (*n* = 6/20)	44% (*n* = 8/18)	48% (*n* = 10/21)
No	50% *(n* = 10/20)	44% (*n* = 8/18)	33% (*n* = 7/21)
Unsure	20% (*n* = 4/20)	11% *(n* = 2/18)	19% (*n* = 4/21)

*n*: the number of total counts collected for the respective category.

## Data Availability

Not applicable.

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
