# Peer review of "The Potential Role of Wild Suids in African Swine Fever Spread in Asia and the Pacific Region"

_viruses, 2022, doi:10.3390/v15010061_

Round 1
Reviewer 1 Report
I can understand the objective of the work and think that a condensed data basis can be of interest especially for decision makers in the region. Nevertheless, I find it most difficult (even after reading it several times) to follow the manuscript and to draw meaningful conclusions from it. This is too mixed-up for me.
The authors should try to straighten the manuscript and to give it a stronger logical thread.
Changes that might improve the manuscript:
· Separate the results of the literature review from the data from the evaluation of the questionnaires.
· The ecology sections contain assumptions regarding the possible role in ASF. This could be reflected in the subheading “Ecology of endemic wild pigs and their potential role in ASF transmission”. Remove statements on ASF occurrence as this should be covered under 3.3 (ASF in wild pigs).
· Remove unnecessary data from tables. There is no need to put the countries that did not report ASF if the heading reads “Reported outbreaks oaf ASF…” Where no data is available, lengthy explanation could be avoided (e.g. the COVID impact). Why give details such as 396 free-ranging pigs?
· I am not sure how to understand the combined information from tables 3 and 4. Example: Indonesia is absent in table 3 despite the fact that in table 4 “domestic and wild swine” are mentioned. The same is true for India, Bhutan and others. What is the discrepancy in wild swine definition?
· I am not sure how to follow section 3.4. Why state “information on distribution, density and ecology … is limited”? The general routes of transmission have been mentioned in other parts and ecology has been dealt with. The same is true for occurrence of ASF. It could be an option to include possible transmission pathways (the documented ones) into the previous section.
· I recommend to restrict the control measures to the ones that are done for wild pigs in particular. How would you implement movement controls in wild pig species? The other information could go into the supplements.
· Avoid duplications.
· Why is table 7 limited to Sus scrofa? Would carcass disposal not apply to other wild suids?
· “Domestic pig farming of Sus scrofa represents are considerable risk for ASFV transmission”? Yes… pigs are the hosts of ASFV. Should we just get rid of all pigs and be done?
· What is the point in describing pig farming in detail when targeting the wild pig role? Change at least the direction… contact is more likely in extensive farming practices…
· If you speak of carcass disposal: Are you referring to domestic pigs or wild pigs? Narrow it down to the relevant parts!
· Sorry, how border controls/ quarantine influence the role of feral pigs in ASF is not clear to me. Again: Narrow it down to relevant parts and explain the relevance for the question at hand.
· Make stronger recommendations in the discussion section and do not focus on uncertainties only.
Minor Points:
I am not really in line with the definition of wild pigs. There is a difference when it comes to true Eurasian wild boar and free-ranging pigs of domestic origin (feral pigs). However, for the purpose of a risk assessment, it may not matter.
Specific points:
· Consider the use of panzootic instead of pandemic (line 37 and throughout the manuscript)
· As the source will never be proven, I suggest adding “probably” (line 64)
· Line 65: Citing 2014 would mean introduction into the European Union (Europe as a continent was affected earlier)
· Line 66: rephrase, it is not completely self-explanatory that you speak of Asia only (remove brackets and include)
· Line 69: cannot follow the sentence, Papua New Guinea is what?
· Line 73 et seq: cannot follow the logic… all statements involve wild boar… please clarify matters
Line 85: typo --> Caucasus
· Line 87 et seq.: restrict the statement to the regions where it is true!
· Line 95 et seq.: cannot completely follow syntax and logic of this sentence
· Line 108 et seq.: Please put the list in order and state the objectives more clearly!
· Line 203: the reference does not completely cover what is said; the study showed rather that the interest in conspecific carcasses was lower than expected. Consider slight revision, put the contact to carcasses in the focus.
· Line 351: strategies
· Line 382: How do you differentiate game and food?
Table 2: Palawan bearded pig
Reviewer 2 Report
Major points
Methods.- Please state the type of review intended, possibly a scoping review with some original work including database insights and online interviews. The search terms or keywords and the search rationale need to be clear and repeatable for anyone performing a similar search later. Explanations given in Lines 143-146 are too vague.
Contribution of wild pigs to ASF transmission.- This is stated as “perceived to be minimal” in the abstract, although also in the abstract it is stated that the Republic of Korea represents an exception. I suggest to put both parts together and re-write the sentences to avoid contradictions. Maybe it is better to focus on the unknown role of wild suids (due to under-reporting or lack of information), rather than on the perception of a minimal role (sometimes potentially policy-driven)?
Transparency in reporting: Reporting wildlife diseases is not at all uniform across countries, and particularly so in several countries of the region of concern for this review. Similarly, ASF reporting in wildlife is almost absent in some European countries (Belarus) and quite restricted for others (Russia). So, the expression in Lines 87-88 “It appears that…” could perhaps be written in a more balanced manner.
ASF control strategies in wild suids: this aspect is insufficiently addressed. I understand this is because there is only limited activity in this regard in the surveyed countries (except Republic of Korea). However, some additional information or discussion, even if only extrapolated from European literature, would be welcome.
Terminology/definitions.- The term “wild pigs” is used throughout the text and defined somewhere later in <line 131. I would prefer to use “wild suids” for a better consistency with current literature. I’d suggest deleting “depending on location” in L100-101. The term wild boar should not be used as a synonym of feral pig.
Minor comments
Line 37v “in certain suids” better than “in suids”
L55 please consider using “Eurasian wild boar (Sus scrofa)” the first time this species is mentioned.
L70 something is missing -- “the only”?
L85 Caucasus
L96 delete “a”
Table 2 please check all names (Babyrus?)
L381-382 a very weak majority if it was an election.
L420 Sus scrofa in italics
L475 specify in warmer regions of… -- carcass survival in cold and temperate regions can be derived from European literature.
Please check references 27 and 34.
Round 2
Reviewer 1 Report
The authors have tried to incorporate almost all suggestions and the manuscript has been improved considerably.
While I would still have preferred some restructuring, I can follow the author's reasoning.
Reviewer 2 Report
I am satisfied with the authors’ answer.